# The Role of Antiviral Prophylaxis in Preventing HBV and HDV Recurrence in the Setting of Liver Transplantation

**DOI:** 10.3390/v15051037

**Published:** 2023-04-23

**Authors:** Sara Battistella, Alberto Zanetto, Martina Gambato, Giacomo Germani, Marco Senzolo, Patrizia Burra, Francesco Paolo Russo

**Affiliations:** 1Gastroenterology and Multivisceral Transplant Unit, Azienda Ospedale-Università di Padova, 35128 Padova, Italy; sarabattistella93@gmail.com (S.B.); alberto.zanetto@unipd.it (A.Z.); martina.gambato@gmail.com (M.G.); marco.senzolo@hotmail.com (M.S.);; 2Department of Surgery, Oncology and Gastroenterology, University of Padua, Via Giustiniani 2, 35128 Padova, Italy

**Keywords:** HBV, HDV, liver transplantation, HBV prophylaxis

## Abstract

Hepatitis B virus (HBV) is a prevalent underlying disease, leading to liver transplantation (LT) for both decompensated cirrhosis and hepatocellular carcinoma (HCC). The hepatitis delta virus (HDV) affects approximately 5–10% of HBsAg carriers, accelerating the progression of liver injury and HCC. The initial introduction of HBV immunoglobulins (HBIG), and then of nucleos(t)ide analogues (NUCs), considerably improved the survival of HBV/HDV patients post-transplantation, as they helped prevent re-infection of the graft and recurrence of liver disease. Combination therapy with HBIG and NUCs is the primary post-transplant prophylaxis strategy in patients transplanted for HBV- and HDV-related liver disease. However, monotherapy with high-barrier NUCs, such as entecavir and tenofovir, is safe and also effective in some individuals who are at low risk of HBV reactivation. To address the problems of organ shortage, last-generation NUCs have facilitated the use of anti-HBc and HBsAg-positive grafts to meet the ever-increasing demand for grafts.

## 1. Introduction

The WHO estimated that approximately 296 million people were affected by chronic hepatitis B (CHB) in 2019, with 1.5 million new infections and 820,000 related deaths per year. The burden of HBV infection is higher in the Western Pacific and African regions, followed by the Eastern Mediterranean region, South East Asia, Europe, and the Americas [1]. The hepatitis delta virus (HDV) is a defective RNA satellite virus that needs the hepatitis B surface antigen (HBsAg) to pass through the cell membrane and infect hepatocytes. HDV may infect along with HBV (coinfection) or subsequently (superinfection), and causes a more aggressive and rapidly progressive form of liver disease. Studies from the 1980s and 1990s reported a 5% prevalence of HDV coinfection among HBsAg carriers [2]. A recent systematic review and meta-analysis estimated that approximately 10.58% of CHB patients suffer from HDV infections in the six continents, excluding Antarctica, and HDV affects 62–72 million people globally [3]. HBV causes several degrees of liver injury, from fibrosis to cirrhosis and its complications, including decompensating events and the development of hepatocellular carcinoma (HCC). An HDV superinfection leads to chronic diseases in over 90% of infected individuals, and cirrhosis occurs in 70–80% of the cases within 5 to 10 years of infection [4]. Nucleos(t)ide analogues (NUCs) have considerably changed the clinical course of liver disease, as they can arrest the progression of hepatic injury and prevent the development of decompensating events. The risk of HCC development is substantially reduced by NUCs, but not completely eradicated, specifically in patients who have already developed advanced chronic liver disease. This finding is supported by the fact that in the last few years, the number of CHB patients transplanted for ESLD has decreased, but the number of HBV-HCC candidates has increased [5]. Historically, HBV and HDV were considered contraindications for liver transplantation (LT) because of the virtually constant reinfection of the graft, which severely affected patient survival. However, the introduction of immunoglobulins (HBIG) and NUCs (in chronological order) significantly modified the disease outcome after LT, making the survival of patients and grafts comparable to those reported for other etiologies. HBV-related liver disease is a major indication for LT, which is the only causal therapeutic option for patients with decompensated cirrhosis and its complications, including acute-on-chronic liver failure and non-resectable HCC. In transplant candidates, antiviral therapy decreases the viral load and, consequently, the risk of re-infection of the graft, whereas after LT, the role of antiviral therapy is to prevent the reinfection of the graft and the recurrence of liver diseases, which can promote long-term patient survival. Several factors contribute to HBV relapse after LT. First, a pre-transplant high viral load is associated with high risk of HBV recurrence following LT; therefore, undetectable HBV-DNA is the aim for all transplant candidates on the waiting list. Second, immunosuppressant therapy for the prevention of graft rejection can increase the risk of HBV reactivation. Third, drug resistance, particularly against lamivudine (LAM) and adefovir (ADV), might develop. Fourth, low compliance of the patient increases the risk of HBV relapse. The gold standard for the management of HBV and HDV patients after LT is the combination of high-barrier NUCs (ETV, TDF, or TAF) with HBIG.

## 2. Prevention of Post-Transplant HBV Recurrence

### 2.1. Definition and Risk of HBV Recurrence

HBV recurrence is defined as the reappearance of HBsAg or HBV-DNA when those markers had previously been undetectable in the same patient. HBV recurrence can result in several degrees of liver damage, including mild self-limited hepatitis, chronic hepatitis, fulminant hepatitis, and fibrosing cholestatic hepatitis [6]. Patients at high virological risk are those with high levels of detectable HBV-DNA (≥4 log) at the time of LT, or those with other virus-associated risk factors, such as HBeAg positivity, a history of antiviral resistance, poor adherence to therapy, HIV coinfection, and those transplanted for acute-on-chronic liver failure resulting from the reactivation of HBV [7]. Patients at low virological risk are those with undetectable viremia or viral load < 4 log at the time of LT and negative for HBeAg, irrespective of the indication for LT. Special populations might be considered as follows: patients with HCC, and patients with HDV coinfection [7]. Antiviral therapy after LT should be tailored to these classes of risk (Table 1).

### 2.2. Hepatitis B Immunoglobulin (HBIG)

HBIG is a human plasma-derived immunoglobulin G concentrate that contains a high titer of neutralizing antibodies of HBsAg (anti-HBs). After HBIG is administered, it forms immune complexes with HBsAg, preventing the infection of hepatocytes. Before 1990, in transplantation cases for HBV-related liver disease, the reinfection of the graft occurred in more than 80% of patients, severely reducing patient survival; the five-year survival was only 50%. In the early 1990s, the use of HBIG was the first attempt to increase the post-transplant survival of HBV patients. The technique revolutionized the post-transplantation outcomes of HBV recipients. Several studies reported higher survival following the administration of HBIG [6,7]. However, the reinfection of the graft was almost imperative after the treatment was stopped. Therefore, different protocols proposed the long-term administration of high doses of HBIG [8,9,10,11]. Monotherapy with HBIG decreased HBV reinfection rates by 19%, as reported in a study in which the patients were followed up for two years. However, in that study, there were no data on duration, dosage, and frequency [11]. A later study proposed high doses of HBIG (10,000 IU) during the anhepatic phase, that is the time between the physical removal of the liver from the recipient and the recirculation of the graft, followed by HBIG daily administration during the first week, and then weekly and monthly administration, to keep the HBsAb levels above 500 IU/L [12]. Subsequent studies showed that lower levels of anti-HBs (e.g., >100 IU/L) could also prevent graft reinfection [13]. HBIG is administered intravenously (IV) during the anhepatic phase and the immediate period after LT to rapidly increase its concentration in the serum. However, for long-term maintenance, IV administration is limited by the need for hospitalization, high costs, and a high rate of complications. Therefore, intramuscular (IM) or subcutaneous (SC) injection is preferred. The administration of HBIG via IM and SC injection can be performed by the patient, and thus it ensures higher compliance with treatment. This technique shows similar long-term results concerning the prevention of HBV reinfection.

### 2.3. Combination of HBIG and NUCs

The long-term use of high doses of HBIG raised the issues of high costs, differences in availability in different countries, parental administration, and ineffective protection against HBV recurrence in some individuals. Thus, after oral antiviral drugs were introduced, HBIG was used in combination with them, which decreased the costs and enhanced the protection against HBV relapse. The introduction of LAM in 1998 significantly enhanced treatment. In the same year, Markowits et al. determined the efficacy of LAM in combination with HBIG in 13 patients in the USA who underwent LT. They found that at a median follow-up of almost one year, HBV recurrence was not observed [14]. Several other studies demonstrated an HBV recurrence rate of less than 10% after 1–2 years of LT in patients treated with a combination of HBIG + LAM [14,15]. However, the use of LAM was soon found to be associated with the development of several viral resistances, which considerably affected the efficacy of post-LT prophylaxis. In a systematic review, Cholongitas et al. showed that ADV in combination with HBIG was associated with a lower risk of HBV recurrence after LT than HBIG + LAM (from 6% to 2%, *p* = 0.024). By using this new combination, the dosage of HBIG could be reduced after the first week of LT [16]. However, the use of ADV was associated with higher costs, renal injury, and a certain risk of viral resistance; thus, it was abandoned in favor of the newer-generation NUCs, such as entecavir (ETV) and tenofovir (TDF). New high-barrier NUCs, i.e., ETV and TDF, can effectively prevent HBV recurrence after LT, and they are safe and well-tolerated [17,18,19]. A systematic review by Cholongitas et al. showed that treatment with high-barrier NUCs combined with HBIG, compared to treatment with the combination of LAM + HBIG, can significantly reduce the HBV relapse rate after LT from 6% to 1%, *p* < 0.001 [20].

### 2.4. Monotherapy Using NUCs

Considering the costs and the need for parental administration of HBIG, several methods have been developed to reduce their application in the last few decades (Table 2). Therapy with only NUCs does not prevent the reinfection of the graft or the establishment of cccDNA in the new hepatocytes. An HBIG-free regimen was first performed with LAM; however, the rate of HBV recurrence was up to 50% within two years of LT because of the development of viral resistance; therefore, that approach was discarded. The introduction of high-potency NUCs, such as ETV and TDF, facilitated the development of cheaper alternative treatments. Fung et al. [21] studied 80 patients who underwent LT for HBV-related liver disease and were treated with ETV monotherapy as recurrence prophylaxis. Although approximately one-fourth of the patients had detectable HBV-DNA levels at the time of LT, most of the transplant recipients showed undetectable levels of HBV-DNA and lost HBsAg at a median follow-up time of 26 months (98.8% and 91%, respectively). No HBV-related death occurred in the cohort; only two patients with detectable HBV-DNA died of sepsis and acute coronary syndrome, respectively. The same research group conducted a study on a cohort of 265 CHB liver transplant recipients treated with ETV monotherapy after LT. They found an overall survival rate of 85% at nine years and an undetectable HBV-DNA rate of 100% at eight years; no death occurred due to HBV recurrence. They also found that the patients who remained persistently positive for HBsAg (14/265) with undetectable HBV-DNA had no significant difference in liver stiffness compared to HBsAg-negative patients (Liver Stiffness Measurement 5.5 vs. 5.2, respectively; *p* = 0.52) [22]. Cholongitas et al. [20] performed a systematic review and found that patients under antiviral prophylaxis with only high-barrier NUCs had a similar HBV-DNA recurrence after LT compared to patients under combination therapy of LAM and HBIG (*p* = 0.14). They also found that the type of NUCs (ETV and TDF) did not influence the HBV recurrence rate. Monotherapy with NUCs is safe and effective for certain patients at low risk of HBV recurrence after LT. However, the long-term effects of the presence of cccDNA or viral integration into the grafts’ hepatocytes need to be determined; therefore, further studies are needed before considering monotherapy using NUCs for all HBV transplant recipients.

### 2.5. Vaccination

Active immunization against HBV has been proposed as an attractive alternative to reduce the medical and financial burden of standard antiviral prophylaxis. Immunosuppressive therapy to prevent the risk of organ rejection after LT leads to a natural reduction of HBV antibodies and a lower response to HBV vaccination [28,29] in terms of seroprotection rate (SPR), which is defined as anti-HBs titers ≥ 10 IU/L. Different techniques were used to improve the SPR; for example, by increasing the dose of vaccine, shortening the time between doses, and adding adjuvants and/or increasing the number of scheduled doses [31,32]. Yang et al. [33] conducted a study in which IM injection of the hepatitis B vaccine was administered at months 0, 1, 2, 6, and 12, co-administering NUCs and HBIG when the titer of anti-HBs was lower than 30 IU/L. All patients had undergone LT at least one year before the study was started and did not have detectable HBV DNA or HBsAg. At the end of the study, one-third of patients responded to active immunization, defined as anti-HBs > 30 IU/L, and no patient experienced HBV recurrence. Responder patients showed lower lymphocytes/eosinophils ratio than non-responder patients; therefore, lymphocytes/eosinophils ratio might be considered to be a helpful criterion for selecting candidates for vaccination. In another study, a three-dose cycle of HBV vaccination led to a SPR in less than 20% of patients transplanted for HBV-related liver disease [34]. HBV vaccination might therefore facilitate the reduction of the dose of HBIG required to prevent HBV recurrence; however, further studies are needed to investigate its safety and efficacy and to determine prognostic factors for response.

### 2.6. Antiviral Prophylaxis Withdrawal

Studies that investigate the feasibility of antiviral prophylaxis withdrawal are limited. The only available prospective study by Lenci et al. [35] followed 30 patients treated with a combination of HBIG and NUCs for at least 5 years after LT. Patients who stopped HBV prophylaxis were strictly monitored every 3–6 months. Six patients experienced HBsAg recurrence, but only three patients required the re-introduction of antiviral prophylaxis. Thus, at a median follow-up time of 6.4 years, 90% of the patients were still prophylaxis-free. Complete withdrawal of antiviral prophylaxis is controversial and is matter of debate also in non-transplanted CHB patients. Recently, Berg et al. [36] conducted a randomized controlled trial, the FINITE study, showing HBsAg loss within 3 years in 19% of CHB patients after cessation of antiviral therapy compared to 0% of patients who continued TDF treatment. However, HBV recurrence may occur in about 60% of patients and hepatic flare in 40% after NUC discontinuation [37,38], and may cause severe consequences, such as liver decompensation and death; therefore, close monitoring is crucial. Complete withdrawal of antiviral prophylaxis after LT might be investigated as a part of clinical trials in low-risk patients who have been transplanted a long time ago, and these patients should be followed up regularly.

## 3. HDV Coinfection

Conventional HBV treatments, such as NUCs, do not affect the HDV viral load. Currently, there is no globally approved treatment for HDV. Pegylated Interferon Alpha (PEG-IFN-alpha) was used for decades to treat HDV infections and showed a virological response in about 25–30% of patients. However, its use is limited by severe adverse effects, which compromised compliance with therapy and its use among patients with decompensated liver disease or in the post-transplant setting. Recently, Bulevirtide, a lipopeptide that blocks the binding of HBsAg-enveloped particles to the sodium taurocholate co-transporting polypeptide (NTCP), which is the cell entry receptor for both HBV and HDV, was approved by EMA at a dose of 2 mg/day in patients with HDV coinfection. Clinical studies confirmed the efficacy and safety of Bulevirtide, even in patients with advanced compensated cirrhosis [39,40,41]. However, several issues still need to be addressed, such as the duration and dosage of antiviral treatment. More studies are needed to understand the mechanisms of the virological response, construct models to predict the therapeutic response, determine the discontinuation criteria, and test the efficacy and safety of the drug in special populations, such as transplanted patients. Two drugs under investigation, PEG-IFN-λ (NCT05070364) and Lonafarnib (NCT03719313), are currently in phase III, but these drugs, as well as Bulevirtide, are not tested on transplanted patients. Specific prophylaxis for patients transplanted for HDV-related liver disease is not available yet; however, since the delta virus needs HBV to infect hepatocytes and replicate, the main objective of the prophylaxis after LT in patients with HDV coinfection is to prevent HBV recurrence. Caccamo et al. compared the effectiveness of HBIG monotherapy to that of the combination of LAM and HBIG and found no cases of HDV recurrence in either group. However, the authors suggested combination therapy as a more cost-effective option, since this strategy allows the use of lower doses of expensive HBIG [42]. Several studies investigated the possibility to discontinue the HBIG therapy in small cohorts of HDV-coinfected patients, showing a recurrence rate of up to 5.8% in a follow-up period from 2 to 20 years [43,44]. The use of NUCs alone rarely leads to HBsAg loss, and this increases the chances of the development of a HDV infection in the new hepatocytes, although the manifestation of a HDV infection is difficult without full reactivation of HBV. However, no effective treatment for HDV infection in the post-transplant setting is known, and the lifelong combination therapy of high-barrier NUCs and HBIG is the recommended prophylaxis [7].

## 4. Hepatocellular Carcinoma

Patients with HCC at the time of LT are at greater risk of HBV recurrence after LT than those without (2–35% vs. 1–9.7%), especially patients at advanced stages of cancer [45]. The exact mechanisms responsible for the increased risk are not fully understood. One explanation could be that extra-hepatic metastatic HCC cells, which are difficult for NUCs to access, might act as a reservoir for the virus; however, this does not explain why HBV can relapse even in the absence of HCC recurrence after LT. Specific prophylaxis for HBV-HCC patients is absent.

## 5. Patients Receiving Anti-HBc-Positive HBsAg Negative Grafts

Despite improvements in perioperative management, mortality on the waitlist remains high [46]. Globally, there is an increasing demand for LT, primarily as a consequence of greater incidence of Non-Alcoholic Steatohepatitis-related cirrhosis and HCC, and, secondarily, as a result of decreased donor suitability, because they are older and have more comorbidities. To meet the growing need for grafts, the use of marginal organs has been implemented, e.g., donation after cardiac death, old age donor grafts, fatty liver grafts, anti-HCV-positive grafts, anti-HBc-positive, and HBsAg-positive grafts. Anti-HBc-positive donors represent a big pool of marginal graft donors, especially in areas with high or intermediate prevalence, such as in Asia and the Mediterranean region, where the prevalence of anti-HBc-positive donors varies from 10% to 50% [47]. Almost invariably, the hepatocytes of anti-HBc-positive donors carry cccDNA, which can transmit the infection in HBsAg-negative recipients. Therefore, these marginal grafts are preferably intended for HBsAg-positive patients, although they are also a feasible option for HBsAg-negative patients, assuming that specific post-transplant prophylaxis is administered (Table 3). A systematic review found that the risk of HBV infection associated with anti-HBc-positive grafts is 48% in naïve patients and 1.4% in those with previous infections (anti-HBc-positive and anti-HBs-positive), suggesting that antiviral prophylaxis is not mandatory in the latter group of patients [48]. Wong et al. [49] conducted a study with almost one thousand patients and found no difference in the graft (76.8% vs. 78.4%) and patient survival (80% vs. 80%) at 10 years from LT in patients who received anti-HBc-positive and anti-HBc-negative grafts. In that cohort, the overall incidence of de novo HBV infection was 2.8%, using antiviral prophylaxis. Only three cases of de novo infection occurred in patients under LAM, while no cases of infection were recorded among patients administered high-barrier NUCs or in anti-HBc-positive and anti-HBs-positive recipients. Several researchers have investigated vaccination for preventing de novo HBV infection, both in children and adults. They found that vaccination is effective if anti-HBs levels are maintained above 100 IU/L [28,37]. Anti-HBc-positive grafts are an essential resource, particularly in highly prevalent regions. The use of antiviral prophylaxis after LT significantly decreased the incidence of de novo HBV infection without affecting the survival of grafts and patients.

## 6. Patients Receiving HBsAg-Positive Grafts

The availability of highly effective antiviral drugs makes the use of HBsAg-positive grafts feasible and safe. The global prevalence of HBsAg-positive donors ranges from 0.5% to 7% among all potential donors, they are relatively common in endemic regions, such as the Far East and the Mediterranean basin [50]. Provided strict selection criteria are used, such as the absence of hepatitis delta infection in donors and recipients and/or the absence of significant fibrosis of the graft, the use of organs from HBsAg-positive donors is associated with favorable outcomes [51,52]. However, a case report showed a good result after LT of a HDV-coinfected patient who received a HbsAg-negative HBV-DNA positive graft. The patient was treated with a combination prophylaxis of HBIG + NUCs and did not manifest HBV-HDV recurrence after LT [53]. HbsAg-positive organs are preferably allocated to HbsAg-positive donors. However, they might also be used in exceptional cases of HbsAg-negative patients with high MELD scores (Table 3). Using a Markov-based model, the use of HbsAg-positive grafts was shown to be cost-effective, particularly for patients with MELD scores above 30, as the costs for life-long use of antiviral drugs were lower than the pre-transplant costs in case of prolonged hospitalization in patients with severely compromised liver function [54]. Ali et al. [55] retrospectively analyzed data from the OPTN database and collected information on 265 patients transplanted with HbsAg-positive grafts from 1999 to 2021; most of these recipients did not have HBV infection before LT (94.7%). No significant difference in patient and graft survival was found between recipients of HbsAg-positive and HbsAg-negative grafts. In that cohort, only one patient who did not have HBV infection before LT died of HBV-related liver disease; the patient was administered LAM as post-transplant prophylaxis. No cases of graft loss were registered among patients treated with high-barrier NUCs, ETV, and TDF. Similar results were recorded in an analysis of data from the China Liver Transplant Registry on 259 HBsAg-negative recipients who received HBsAg-positive grafts and were compared to those who received HBsAg-negative grafts, with one-year survival rates of 78.92% vs. 85.65%, respectively, and five-year survival rates of 58.1% vs. 69.1%, respectively; log-rank *p* = 0.06 [56]. In this study, patients who received HBsAg-positive grafts had higher rates of HBV recurrence, but this might be explained by the fact that almost half of the recipients were treated with LAM as an antiviral drug. No conclusive results were available on the risk of HCC after LT in recipients of HBsAg-positive grafts. In the retrospective study by Ali et al. [55], where data from the OPTN registry were used, more than 20% of data on the incidence of post-transplantation malignancy were missing. The risk is assumed to be higher among those receiving grafts from donors who acquired HBV infection perinatally. Treatment using NUCs can reduce the risk of HCC, particularly in patients without advanced liver disease. Further studies on the risk of HCC in patients transplanted with HBsAg-positive grafts and precise indications for post-transplant surveillance are needed.

## 7. Conclusions

The hepatitis B and delta viruses are still common underlying diseases leading to LT, especially for HCC. The introduction of HBIG and high-barrier NUCs significantly modified the outcomes after LT. Their application in the post-transplant setting improved graft and patient survival, making them comparable to those reported in other indications for LT. The main treatment strategy for the prevention of HBV recurrence after LT is the combination of HBIG and NUCs; however, HBIG may be stopped based on an individualized model that considers risk factors for recurrence (e.g., HDV, HCC), feasibility (e.g., patient compliance), and medication costs, in the setting of clinical trials. Because highly effective and safe antiviral drugs are available, the use of anti-HBc and HBsAg-positive grafts is viable, preferably addressed to HBV-infected patients or uninfected patients with severely compromised liver function. Further studies with larger cohorts are needed to develop better protocols for individualized prophylaxis.

## Figures and Tables

**Table 1 viruses-15-01037-t001:** Risk of HBV recurrence after LT.

Risk of HBV Recurrence	
Low	HBV-DNA undetectable or <4 log at the time of LT HBeAg negativity at the time of LT Good compliance with antiviral therapy
High	HBV-DNA ≥ 4 log at the time of LT HBeAg positivity at the time of LT History of antiviral resistances Poor adherence to antiviral therapy HIV coinfection Acute-on-chronic liver failure as an indication for LT
Special population	HCC HDV coinfection

Abbreviations: LT, liver transplantation; HCC, hepatocellular carcinoma.

**Table 2 viruses-15-01037-t002:** Published studies in which HBIG-free prophylaxis was administered to prevent HBV recurrence after LT.

	Number of Patients	Type of Antiviral Drug	Use of HBIG	HBV Recurrence Rate	Follow-Up Time
Radhakrishnan et al. [23]	42	NA	HBIG discontinuation after 5 days from LT	2% HBV-DNA transitory recurrence 2.9% HBsAg recurrence	3.1 years
Saab et al. [24]	79	ETV, TDF, LAM, TAF or ADV (monotherapy or combination)	HBIG discontinuation	13.9% HBsAg recurrence	6.5 years
Fung et al. [21]	80	ETV monotherapy	NO	98.8% HBV-DNA undetectable 91% HBsAg undetectable	26 months
Tapermann et al. [25]	18	TDF (plus fixed-dose emtricitabine)	HBIG discontinuation	100% HBV-DNA and HBsAg undetectable	72 weeks
Manini et al. [26]	77	ETV or TDF	HBIG discontinuation	100% HBV-DNA undetectable 91% HBsAg undetectable	5 years
Cholongitas et al. [27]	47	LAM + ADV, LAM + TDF, TDF, ETV	HBIG discontinuation	6.3% HBsAg recurrence, with undetectable HBV-DNA	24 months
Wadhawan et al. [28]	75	LAM + ADV, TDF, ETV	NO	100% HBV-DNA undetectable 88% HBsAg undetectable	21 months
Muthiah et al. [29]	35	ETV or TDF	NO	87.9% HBV-DNA undetectable 72.4% HBsAg undetectable	80 months
Fung et al. [22]	265	ETV	NO	100% HBV-DNA undetectable 92% HBsAg undetectable	59 months
Fung et al. [30]	362	LAM, ETV, combination therapy (predominantly LAM + ADV)	NO	98% HBV-DNA undetectable 88% HBsAg undetectable	53 months

Abbreviations: LAM, lamivudine; ADV, adefovir; TDF, tenofovir; ETV, entecavir; HBIG, hepatitis B immunoglobulins.

**Table 3 viruses-15-01037-t003:** Indications for allocation and therapeutic/prophylactic recommendations of anti-HBc and HBsAg-positive grafts.

Donor	Recipient				
	HBsAg+	HBsAg+ HDV+	HBsAg− Anti-HBc+ Anti-HBs+	HBsAg− Anti-HBc− Anti-HBs−	HBsAg− Anti-HBc− Anti-HBs+
**HBsAg+**	High-barrier NUCs	Not recommended	High-barrier NUCs	Only in selected cases (e.g., very high MELD) High-barrier NUCs	Only in selected cases (e.g., very high MELD) High-barrier NUCs
**Anti-HBc+**	High-barrier NUCs + HBIG	High-barrier NUCs + HBIG	No prophylaxis	High-barrier NUCs	High-barrier NUCs

Abbreviations: NUCs, nucleos(t)ide analogs; HBIG, HBV immunoglobulin; MELD, model for end-stage liver disease.

## Data Availability

Not applicable.

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
