# Peer review of "The Role of Antiviral Prophylaxis in Preventing HBV and HDV Recurrence in the Setting of Liver Transplantation"

_viruses, 2023, doi:10.3390/v15051037_

Round 1

Reviewer 1 Report

Well written review paper on HBV and HDV recurrence following Liver Transplantation. It is pretty comprehensive and the references are appropriate and mostly recent. 

- Minor Change recommended for the Title; "HBV and HDV" suggested instead of "HBV-HDV'.  One may assume this is a specific review for HBV HDV coinfected transplanted  cases only  but this is rather a review on HBV as well as HDV cases.

- HDV liver transplant is rather unusual in North America. There is case report in ACG Journal from 2021, talking about the importance of HBIG in such cases. It can be considered as a supplemental reference to Section 3. HDV Coinfection

doi: 10.14309/crj.0000000000000582

Author Response

We thank the reviewer for his/her useful comments and suggestions. The manuscript has been revised as requested.

  • We use HBV and HDV in the title as suggested
  • We cited the case report of the HBV/HDV patient transplanted with a HBsAg-positive HBV-DNA-negative graft in the paragraph “Patients receiving HBsAg-positive grafts

Reviewer 2 Report

Overall, this review is a good addition to the field and may be helpful for different groups of readership. However, several statements are misleading/easy to misunderstand at best, and factually false at worst. Therefore, they need to be revised. Also, some points could be added to be discussed.

Individual comments are in the attached document.

English language is good, however some sentences and paragraphs are written in a confusing way and need editing.

Author Response

We thank the reviewer for his/her useful comments and suggestions. The manuscript has been thoroughly revised.

Major comments:

 Abstract:

  1. HBV is not the indication, but the underlying cause for LT. Cirrhosis and HCC are the indication.
  2. “…post-transplant prophylaxis strategy in HBV/HDV recipients” – what does that mean? Do you mean recipients of HBV/HDV-infected livers? HBV/HDV-positive recipients of LT?

1.We replaced the terms as indicated

2.We meant the post-transplant antiviral prophylaxis in patients who undergo LT for HBV and HDV related liver disease. We corrected the sentence in order to make them more comprehensible.

 Introduction:

  1. HDV is not RNA-defective. It is a defective RNA satellite virus.
  2. HBIG and NAs did not modify the natural history (do you mean course?) of disease. When there’s treatment, it is not anymore the natural course of disease. Suggestion: replace with: modified disease outcome after LT.
  3. HBV alone is never an indication for LT (see comment for abstract). Needs to be worded differently.
  4. … LT is not the only therapeutic option, but the only causal therapeutic option.
  5. “therefore, virtually all transplant candidates on the waiting list have undetectable HBV-DNA” – causal statement is not true. Suggestion: replace with: therefore, undetectable HBV-DNA is aimed for in all transplant candidates on the waiting list.

1.We modified the definition of HDV, as suggested

2.We replaced the term “history” with “course”, as suggested

3 and 4.We corrected the sentences as indicated

5.We followed the suggestion for the sentence

 Section Prevention of post-transplant HBV recurrence:

Subsection Definition and risk of HBV recurrence:

  1. Patients were likely not “initially freed from the virus”. Rephrase, suggestion: reoccurrence of HBsAg or HBV-DNA when those markers had previously been undetectable in the same patient.
  2. “The risk of HBV recurrence after LT can be classified as high and low risk. “ – This sentence does not add any information.
  3. “or those with other markers of high viral load” – either misleading or inaccurate. Suggestion: replace with: or those with other virus-associated risk factors.
  4. Special populations: “patients at risk of poor compliance with the antiviral therapy after LT” – this population was already mentioned above in the high-risk group, and also does not appear under “special populations” in the table; and should therefore not be added here again.

1.We rephrased the sentence

2.We deleted the sentence

3.We replaced the sentence, as suggested.

4.We eliminated “patients at risk of poor compliance” from the special populations group.

 Subsection Hepatitis B immunoglobulin (HBIG):

  1. “the reinfection of the graft was almost constant after the treatment was stopped” – suggest to replace constant with imperative for better comprehensibility.
  2. ..decreased HBV reinfection rates by 19%
  3. “However, in that study, there were no data on duration, dosage, and frequency. In the first few years of post-transplant monoprophylaxis treatment, HBIG was administered at high doses (10000 IU) during the anhepatic phase, followed by daily administration during the first week, and then weekly and monthly administration, to keep the HBsAb levels above 500 IU/L.” – First you claim there’s no data on dosage, but then you list data on dosage. This is confusing – recommend to make clear that the first sentence applies to the cited study, and the second (likely?) to general best practice at the time. You should also explain to readers what the anhepatic phase is.

1.We replaced with imperative as suggested

2.We corrected the sentence

3.We clarified the sentences and explained what anhepatic phase is

Subsection Combination of HBIG and NAs:

  1. “and ineffective treatment of some individuals” – unclear what is meant by this?
  2. “Several other studies demonstrated 117 an HBV recurrence rate of less than 10% after 1–2 years of LT” – are those for LAM alone or LAM + HBIG?
  3. “in favor of the last-generation NAs” – suggest to replace last-generation with newer-generation.

1.We deleted the sentence

2.The study was referred to LAM + HBIG, we specified in the manuscript

3.We replaced the term last generation with newer generation

Subsection Monotherapy using NAs:

  1. “the seropositivity for HBV has no significant clinical consequence if viral suppression is maintained” – need to define which marker for seropositivity you are referring to here. Also, this statement is very questionable (clinical consequences are, for example, also considerations of risk of infectivity with regards to other people etc.), and later in the paragraph it is contradicted by “the long-term effects of the presence of cccDNA or viral integration into the grafts’ hepatocytes need to be determined”, so suggest strongly to rephrase.
  2. “No HBV-related mortality occurred in the cohort” – suggest to replace mortality with death.

1.We rephrased the sentence

2.We replaced “mortality” with “death”, as indicated

Subsection Vaccination:

  1. For comprehensibility, it would be helpful to add 2-3 sentences at the beginning of this paragraph on the overall concept of vaccination after LT in these patients and why it is done. Here, you should also define/explain what you mean by “seroprotection rate” and “protective seroconversion” which are terms used later in the paragraph.
  2. “…when the title of anti-HBs was lower….” titer, not title.
  3. “of HBV transplanted patients” – what does that mean? HBV-infected patients after LT?
  4. “identify the best candidate” – if you say this, you should also mention the different candidates and explain what is currently used as best practice.

1.We added some sentences at the beginning of the paragraph to better explain the rationale of vaccination and we defined the “seroprotection rate”, replacing the term “protective seroconversion” with “seroprotection rate” in order not to confuse the reader

2.We corrected the sentence

3.We meant patients transplanted for HBV related liver disease; we specified in the test

4.We eliminated the part of the sentence

Subsection Antiviral prophylaxis withdrawal:

  1. Here, you could refer to recent studies on NA discontinuation in non-transplanted patients and discuss that even this is still a debatable topic.

1.We briefly discussed the topic of NUCs discontinuation in non-transplanted patients.

Section HDV coinfection:

  1. “its use is burdened by severe adverse effects” – suggest to replace burdened with limited.
    1. “Bulevirtide, an NTCP inhibitor, showed promising results in phase II and ongoing phase III trials” a. Bulevirtide is not an NTCP inhibitor, strictly speaking. It binds to NCTP and competitively blocks virus entry.
    2. Should explain that NTCP is the bona fide receptor for HBV/HDV entry.
    3. There are now many real-world data, since it has been conditionally approved by EMA and in use in Europe for a few years. Suggest to rephrase: promising results in HBV/HDV co-infected patients, especially on HDV infection and liver damage.
    4. “and its possible association with Peg IFN-alpha” – What do you mean by that? Mechanisms of interplay? Please rephrase, as otherwise it would be unclear, since there are already studies published on Bulevirtide/interferon combination therapy.
    5. “Other drugs under investigation are currently in phase Ib/IIa, but these drugs still need to be tested on transplanted patients.” When this is mentioned, you should also mention which drugs you are referring to. This also suggests that Bulevirtide, by contrast, has already been tested in transplanted patients? Please either rephrase or provide a reference.
    6. “in patients with HDV is to prevent HBV infection” – suggest to replace infection with recurrence, to prevent misunderstandings.
    7. “Caccamo et al. compared the effectiveness of HBIG monotherapy to that of the combination of LAM and HBIG and found no cases of recurrence in either group.” – please indicate which virus this study refers to.
    8. “However, the authors suggested combination therapy as a more cost-effective option” – please explain to the readers why, as since this sounds counter-intuitive.
    9. “Several studies proposed that HBIG therapy should be discontinued in small cohorts of HDV-coinfected patients. The patients were followed up for 2 to 20 years, and they showed a recurrence rate of up to 5.8%. The use of NAs alone made it hard to detect HBsAg, which increased the chances of the development of an HDV infection in the new hepatocytes, although the manifestation of an HDV infection is difficult without full reactivation of HBV.” – This entire part needs proofreading/rephrasing as it is difficult to understand and easy to misunderstand, especially for readers unfamiliar with the topic.
    10. “However, no effective treatment for HDV infection is known” – Questionable, since Bulevirtide – and, in subgroups of patients, interferon – seems to be an effective treatment for HDV. Please rephrase, and indicate if you are only referring to LT patients here or all patients.

1.We replaced the term “burdened” as indicated

2.We specified the activity of BLV

3.We eliminated that sentence

4.We specified which are the drugs under investigation and cited the numbers of trial

5.We corrected as indicated

6.We specified that patients were HBV and HDV coinfected

7.We explained why combination therapy is more cost effective (it allowed a dose reduction of HBIG)

8.We rephrased the entire sentence to make it more comprehensible

9.We specified that we were referring to the post-transplant setting

Section Hepatocellular Carcinoma

  1. It may be worth to also refer to investigations of the role of viral integrations into the host genome here.

1.We did not investigate the role of viral integration into the host genome since this is out of the topic of our manuscript

Section Patients receiving anti-HBc-positive grafts

  1. Please clarify in the headline that this section is about anti-HBc-positive, but HBsAg-negative grafts (in contrast to the following section).
  2. “the use of marginal organs has been implemented” – define what you mean by “marginal organs”, or rephrase?
  3. “The hepatocytes of anti-HBc-positive donors carry cccDNA” – in general yes, but not in (rare) cases of actual cure (e.g. possibly when infection is resolved after acute infection) – please rephrase.
  4. “no difference in the graft (76.8% vs. 78.4%) and patient survival even after 10 years of LT (80%)” – which percentage refers to what? 76.8%/78.4% at what time point? 80% after ten years? For both groups? Please clarify.
  5. The shortage of organs is mentioned a few times in this and the following section. It would be nice to actually refer to some data, e.g. exemplary from some regions on how many organs are needed/provided/how long patients wait/what mortality is on the waiting list, to underline that there is actually a shortage.

1.We clarified the title

2 and 5. We shortly discuss the topic of organ shortage and marginal grafts

  1. We rephrased as suggested

4.We rephrased the sentence to make it clearer

Section Patients receiving HBsAg-positive grafts

  1. “The shortage of organs and the availability of highly effective antiviral drugs make the use of HBsAg-positive grafts safe” – shortage of organs doesn’t make the use of those grafts safe (antiviral drugs, monitoring etc., however, may). Please rephrase.
  2. “such as the absence of hepatitis delta infection in donors and recipients of and/or graft significant fibrosis, are implemented.” – this part is very confusing, please rephrase/clarify.
  3. “The prevalence of HBsAg-positive donors ranges from 0.5% to 7% among all potential donors” – worldwide? Across which regions?
  4. “collected information on 265 transplanted HBV-positive grafts from 1999 to 2021; most of these grafts did not have HBV infection before LT” – please clarify how many of those were HBsAg-positive?
  5. “No significant difference in patient and graft survival was 272 found between recipients of HBV-positive and HBV-negative grafts.” – Is HBV-positive here the same as HBsAg-positive, or was a different marker used?
  6. “one-year survival rates of 78.92% vs. 85.65%” it would be very interesting to know the p-value for these two numbers, especially because log-rank p = 0.06 for the following comparison approaches significance. This may also be worth a comment, that the outcome is close to significant, which opens room for further discussion.
  7. “recipients of HBsAg grafts” – probably mean HBsAg-positive grafts.
  8. and 2. We rephrased as indicated

3.Worldwide, we specified

4.94.7% of patients were HBV negative, we specified in the manuscript

5.We modified HBV positive with HbsAg positive

6.We discussed the results as suggested

7.We meant transplanted patients who received HbsAg-positive grafts

Table 3: the headline doesn’t really give information of the table content, suggest to include indication of therapeutic/prophylactic recommendations into the title.

We modified the headline to give more information of the table content

Section Conclusions:

  1. “The hepatitis B and Delta viruses are still common indications for LT” – see comments above. Viruses are not indications for LT. Liver failure/cirrhosis/HCC is.
  2. “The introduction of HBV immunoglobulins” – Suggest to use the term HBIG here to be consistent with previous paragraphs.
  3. “significantly modified the history of liver disease” – suggest to rephrase, e.g. significantly modified outcomes….
  4. “risk factors for recurrence and medication costs” – suggest to add feasibility here (compliance etc.)
  5. “to HBV patients or severely compromised non-HBV patients” – suggest to rephrase: HBV-infected patients or uninfected patients with severely compromised liver function.

1.We corrected as above

2.We used the term HBIG

3.We used the term outcomes

4.We add feasibility, as suggested

5.We rephrased as indicated.
